# Agroecological Zone-Specific Diet Optimization for Water Buffalo (*Bubalus bubalis*) through Nutritional and In Vitro Fermentation Studies

**DOI:** 10.3390/ani14010143

**Published:** 2023-12-31

**Authors:** Sultan Singh, Pushpendra Koli, B. P. Kushwaha, Uchenna Y. Anele, Sumana Bhattacharya, Yonglin Ren

**Affiliations:** 1ICAR-Indian Grassland and Fodder Research Institute, Jhansi 284 003, India; singh.sultan@rediffmail.com; 2College of Environmental and Life Sciences, Murdoch University, 90 South Street, Murdoch, WA 6150, Australia; 3ICAR-Central Institute for Research on Buffaloes, Hisar 125 001, India; bpkush64@gmail.com; 4Department of Animal Sciences, North Carolina Agricultural and Technical State University, Greensboro, NC 27411, USA; uyanele@ncat.edu; 5Natcom Management Cell, Ministry of Environment and Forests, New Delhi 110 003, India; sumana_bhattacharya@yahoo.com

**Keywords:** agroecological zone, buffalo, diet formulation, feeding system, fermentation, methane emission

## Abstract

**Simple Summary:**

This study involves the formulation of distinct diets for water buffalo based on locally available feed resources to specific agroecological zones. The diets were categorized into three groups addressing the maintenance, growth, and lactation/production requirements of buffaloes. This study assessed the chemical composition and in vitro gas and methane emissions of each diet. The implication of this work suggests a promising future for buffalo feeding systems, as it focuses on need-based formulations using specific regional ingredients. This approach may enhance the efficiency and sustainability of buffalo farming in specific zones.

**Abstract:**

The water buffalo faces challenges in optimizing nutrition due to varying local feed resources. In response to this challenge, the current study introduces originality by addressing the lack of region-specific feeding strategies for water buffaloes. This is achieved through the formulation of 30 different diets based on locally available resources, offering a tailored approach to enhance nutritional optimization in diverse agroecological contexts. These diets were segmented into three groups of ten, each catering to the maintenance (MD_1_ to MD_10_), growth (GD_1_ to GD_10_), and lactation/production (PD_1_ to PD_10_) needs of buffaloes. Utilizing local feed ingredients, each diet was assessed for its chemical composition, in vitro gas and methane emissions, and dry matter (DM) disappearance using buffalo rumen liquor. The production diets (127 and 32.2 g/kg DM) had more protein and fats than the maintenance diets (82.0 and 21.0 g/kg DM). There was less (*p* < 0.05) fiber in the production diets compared to the maintenance ones. Different protein components (P_B1_, P_B2_) were lower (*p* < 0.05) in the maintenance diets compared to the growth and production ones, but other protein fractions (P_B3_, P_c_) were higher (*p* < 0.05) in the maintenance diet. Furthermore, the growth diets had the highest amount of other protein components (P_A_), while the maintenance diets had the highest amount of soluble carbohydrates (586 g/kg DM), whereas the carbohydrate fraction (C_B1_) was highest (*p* < 0.05) in the production diets (187 g/kg DM), followed by the growth (129 g/kg DM) and maintenance diets (96.1 g/kg DM). On the contrary, the carbohydrate C_A_ fraction was (*p* < 0.05) higher in the maintenance diets (107 g/kg DM) than in the growth (70.4 g/kg DM) and production diets (44.7 g/kg DM). The in vitro gas production over time (12, 24, and 48 h) was roughly the same for all the diets. Interestingly, certain components (ether extract, lignin, NDIN, ADIN, and P_B3_ and C_C_) of the diets seemed to reduce methane production, while others (OM, NPN, SP, P_A_ and P_B1,_ tCHO and C_B2_) increased it. In simple words, this study reveals that different diets affect gas production during digestion, signifying a significant step towards a promising future for buffalo farming through tailored, region-specific formulations.

## 1. Introduction

India’s vast agroecological diversity offers a surplus of locally available feed resources. These diverse regions provide an opportunity to create diets specific to the needs of buffaloes, depending on their lifecycle stage, be it maintenance, growth, or lactation. The country’s agricultural backbone stands not just on its crops but, significantly, also on its livestock, with buffaloes playing a pivotal role [1]. The buffalo, often deemed the ‘Black Gold’ of India and Pakistan, is central to the rural economy [2,3]. This is not surprising given that India boasts a multitude of buffalo breeds, each with its unique attributes, suiting the varied climatic and topographical conditions of the country. From the *Murrah*, known for its high milk yield, to the *Bhadawari*, appreciated for its adaptability, the diversity is truly expansive [4]. In India, buffalo and cattle farming face challenges with limited feed resources and insufficient farmer knowledge on animal nutrition, impacting dairy animal productivity, which varies across regions due to differences in feed availability, types, and adherence to scientific feeding practices [5]. Livestock is the primary contributor of 50% of the 14.17 Tg methane emission total that comes from the Indian agricultural sector [6]. Methane, an important GHG (greenhouse gas) about 22–25 times more potent than carbon dioxide [7], is produced by ruminants as an end product of microbial digestion [8]. During the fermentation process (digestion and metabolism) of diets in the gastrointestinal tract, between 2 and 12% of dietary gross energy is lost as methane [7]. Several factors, viz., animal species and size, animal physiological stage, feed intake, digestibility, diet composition, etc., influence enteric methane production [9,10]. Diet composition (chemical and physical qualities) and its intake level (quantity consumed) influence methane production due to their effect on the rate of digestion and the rate of passage [11]. Animal species and diet composition play an important role in methane production [7,12,13]. Animals have three main nutritional needs: staying healthy (maintenance), growing, and producing things like milk or offsprings (production). To meet these needs, we created three different diets for each region. These diets were made by combining different amounts of locally available food resources, like dry and green roughages, along with concentrated mixtures. Protein in vitro fermentation has been shown to be associated with lower CH_4_ production than carbohydrates fermentation [14,15]. Dietary nitrogen (N) concentrations play an important role in influencing rumen methanogenesis [16], specifically where feed N is low [17], leading to the reduced microbial growth of methanogens, which face difficulty competing under low N conditions [18]. The type of carbohydrate being digested, such as cellulose, hemicelluloses, and soluble residue, holds a notable influence over methane production [19,20,21]. Moreover, a strong relationship is observed between CH_4_ production and digestible neutral detergent fiber (NDF) for cows and calves [22]. 

The main goal of this study was to develop and evaluate 30 water buffalo diets tailored for various life stages and agroecological zones in India. The assessment involved scrutinizing their nutritional compositions and in vitro methane production. The ultimate aim was to redirect methane emissions into a valuable energy source, thereby improving livestock productivity and simultaneously addressing global environmental concerns.

## 2. Materials and Methods 

### 2.1. Formulation of Concentrate Mixtures 

Local feed ingredients and their use in feeding livestock according to their suitable agroecological regions were considered for the formulation of concentrate mixtures (CM) for different regions of the country. A total of nine CM were prepared using protein and energy sources in different proportion for use in different diets as described in Table 1.

### 2.2. Preparation Diets/Rations

The nutritional requirements of livestock were classified into three categories based on animals’ functional needs, viz., maintenance, growth, and production. For each category, ten diets/rations were prepared, and, hence, a total of thirty diets were formulated via the uniform mixing of dry and green fodder with concentrate mixtures in different proportions (Table 2).

### 2.3. Determination of Chemical Composition 

The dry matter (930.15), ash (932.05), N (976.05), and ether extract (EE, 920.39) contents of the diets’ samples were estimated following the standard method of the Association of Official Analytical Chemists (AOAC) [23]. The nitrogen values were multiplied by 6.25 to convert them into crude protein (CP) values. Neutral detergent fiber (NDF), acid detergent fiber (ADF), cellulose, and lignin (sa) were estimated as per the sequential method [24] using a fiber analyzer (Fibra Plus FES 6, Pelican, Chennai, India). Both the NDF and the ADF were expressed inclusive of their residual ash. There was no complex plant matrix included in our diet compositions; therefore, heat-stable α-amylase and sodium sulfite were not used in NDF determination. The lignin (sa) was determined by solubilizing cellulose with 72% of sulfuric acid in the ADF residue [24]. The cellulose was calculated as the difference between the ADF and the lignin (sa) in the sequential analysis. The hemicellulose was calculated as the difference between the NDF and ADF contents.

### 2.4. Estimation of Carbohydrate Fractions

The carbohydrate (CHO) fractions of the different diets samples were estimated as per the Cornell Net Carbohydrate and Protein (CNCP) system [25]. This system classifies CHO fractions into four fractions, as follows: C_A_ indicates rapidly degradable sugars; C_B1_ classifies intermediately degradable starch and pectin; C_B2_ includes slowly degradable cell walls; and C_C_ comprises unavailable/lignin-bound cell walls based on their degradation rate. The diets’ total CHO (tCHO; g/kg DM) was determined by subtracting the CP, ether extract (EE), and ash contents from 1000. The structural carbohydrates (SC) were calculated as the difference between the NDF and the neutral detergent-insoluble protein (NDIP), and the non-structural CHO were estimated as the difference between the tCHO and the SC [26]. For the starch estimation, samples were extracted with ethyl alcohol to solubilize free sugars, lipids, pigments, and waxes. The residue rich in starch was solubilized with perchloric acid, and the extract was treated with anthrone–sulfuric acid to determine glucose colorimetrically using a UV spectrophotometer (LABINDIA3000) at 630 nm [27].

### 2.5. Estimation of Protein Fractions

The CP fractions of the diets were partitioned into five fractions according to the CNCPS, [25] as modified previously [28]. These are the following: fraction P_A_, indicating non-protein N, estimated as the difference between the total N and the true CP N precipitated with sodium tungstate (0.30 M) and 0.5 M of sulfuric acid; P_B1_, the buffer-soluble protein calculated as the difference between the true protein and the buffer-insoluble protein, estimated with a boratephosphate buffer (pH 6.7–6.8) and a freshly prepared 0.10 M sodium azide solution; fraction P_B2_, the neutral detergent-soluble protein, estimated as the buffer-insoluble protein minus the ND-insoluble protein; fraction P_B3_, the acid detergent-soluble CP, estimated as the difference between the ND-insoluble protein and the acid detergent-insoluble CP; and fraction P_C_, assumed to be indigestible (protein associated with lignin, tannin–protein complexes, and Maillard products which are unavailable to animals).

### 2.6. In Vitro Incubation

The in vitro gas production was determined using the pressure transducer technique [29]. Ruminal fluid was collected before feeding from two fistulated adult male *Murrah* water buffaloes (*Bubalus bubalis*) fed a wheat straw-concentrate diet (65:35 DM basis). The rumen fluid was filtered through a double layer of cheese cloth and bubbled with CO_2_ before the commencement of incubation. The incubation medium was prepared by means of the sequential mixing of a buffer solution (NH_4_HCO_3_ and NaHCO_3_), a macro-mineral solution, a micro-mineral solution, and resazurin solution [30]. Samples (1 g) of air-dry green forages were weighed into three serum bottles (150 mL capacity). Three serum bottles without substrate were used as blank cultures. The sample and control serum bottles were gassed briefly with CO_2_ before adding 65 mL of medium. The bottles were continuously fluxed with CO_2_, and then 3 mL of reducing solution were added in each bottle. The gassing of bottles with CO_2_ continued until the pink color turned colorless. Before inoculation, the gas pressure transducer was used to adjust the head-space gas pressure in each bottle (to adjust the zero reading on the LED display). The serum bottles were inoculated with 8 mL of ruminal fluid inoculum using a 10 mL syringe. The inoculated bottles were sealed and incubated at 39 °C. The samples were incubated in triplicates, and the gas production (mL) was measured at 12, 24, and 48 h of incubation. The whole process was repeated on a different day.

### 2.7. Methane Measurements

The methane (CH_4_) in the total gas was measured from three bottles incubated for each of the thirty diets at the 12, 24, and 48 h timepoints using gas chromatography (Nucon 5765 microprocessor-controlled gas chromatograph (GC), Okhla, New Delhi, India) equipped with a stainless-steel column packed with Porapak-Q and a flame ionization detector. Gas (1 mL) was sampled from the gas produced using a Hamilton syringe and injected manually (pull and push method of sample injection) into a GC. The GC was calibrated using standard methane and CO_2_. The methane level was additionally measured in blank samples at different fermentation stages, and these measurements were used to correct for methane produced by the inoculum. The methane measured was related to the total gas to estimate its concentration [31] and converted into energy and mass values using 39.54 kJ/L CH_4_ and 0.716 mg/mL CH_4_ factors, respectively [32].

### 2.8. In Vitro Dry Matter Digestibility (IVDMD) and Energy of Diets

For the determination of the IVDMD for the evaluated diets a standard method was followed [33], wherein a 0.5 g sample was incubated for 48 h and then digested with 0.1 g of pepsin (1:3000 Sisco Research Laboratories, Mumbai, India) and 2 mL of 6N HCl at 39 °C for 24 h. The samples were incubated in triplicate with ruminal inoculum from the two fistulated buffaloes described previously. A provision was also made for the blanks, as described for the in vitro gas production. The digestibility was estimated as the difference between the DM incubated and the residual DM at the end of the second stage of digestion. The gross energy (GE) of the forages was measured with a bomb calorimeter (Toshniwal Brothers CLOI/M2, Bangalore, India) using benzoic acid as the standard. 

### 2.9. Statistical Analysis

The data were subjected to an analysis of variance using the GLM procedure of SAS (2002). The model was the following: Yij = [1] + Fi + Eij, where Yij represents the individual observation of the variable, and Fi is the fixed effect of the ith diet (i = 1–30). The overall mean is expressed as [1], and Eij is the random error associated with Yij, not accounted in the fixed effect. The means were separated using Fisher’s LSD and all the statistical tests were at the *p* = 0.05 level of significance. The means of cereals, grasses, and legumes were compared using orthogonal contrasts (i.e., cereals vs. grasses, cereals vs. legumes, and grasses vs. legumes). The differences among forage means with *p* < 0.05 were accepted as statistically significant. A correlation analysis was used to establish relationships between chemical constituents, carbohydrate fractions, protein fractions, and CH_4_ production. Pearson’s correlation analysis was performed to establish the relationship of chemical composition with methane production, carbohydrate fractions, and protein fractions at level *p* < 0.05.

## 3. Results

### 3.1. Chemical Composition

The crude protein (CP) and ether extract (EE) values were significantly higher (*p* < 0.05) for the production diets (127 and 32.2) than the maintenance diets (82.0 and 21.0 g/kgDM), respectively. The CP values of all three diets including the MD, the GD, and the PD varied, measuring 69.8–96.1, 106–130, and 103–153 g/kg DM, respectively (Table 3), whereas, the concentrations of NDF, ADF, and cellulose were lower (*p* < 0.05) in the production diets (546, 333 and 245) than in the maintenance diets (618, 395 and 293 g/kg DM). Interestingly, no significant difference was observed in the lignin contents among all three diets. 

### 3.2. Nitrogen Fractions

The protein fraction values (P_B1_ and P_B2_; g/kg DM) followed a lower to higher (*p* < 0.05) order from the MD (150.2 and 357.3) to the GD (204.7 and 409.7), followed by the PD (217.4 and 412.3), while the mean concentration of the slowly degradable protein fraction (P_B3_) and the lignin-bound protein fraction (Pc) were higher (*p* < 0.05) in the maintenance diets (205.6 and 181.3) than in the growth diets (113.9 and 114.8), followed by the production diets (151.5 and 104.0 g/kg DM) (Table 4). The average value of P_A_ (g/kg DM) was significantly higher (*p* < 0.05) for the growth diets (136.9) than in the production and maintenance diets, which had values of 114.8 and 105.6, respectively. In the maintenance diets, the affinity of protein binding to ADF was observed to be higher than in the growth and production diets, whereas the SP concentration was in the reverse order, meaning that the concentration in the maintenance diets was lower than in the two other diets.

### 3.3. Carbohydrate Fractions

Among the carbohydrate fractions, no significant difference was observed among all three diets for the total carbohydrate levels (tCHO), while the SC contents were (*p* < 0.05) higher in the maintenance diets (586.2) than in the production diets (513.0 g/kg DM), respectively. The average value of the rapidly degradable carbohydrate fraction (C_B1_) differed (*p* < 0.05) among the diets, being highest in the production diets (187.2), followed by the growth (129.5) and maintenance diets (96.1 g/kg DM; Table 5). The contrary carbohydrate C_A_ fraction was (*p* < 0.05) higher in the maintenance diets (107.1) than in the growth (70.4) and production diets (944.7 g/kg DM). The carbohydrate fractions C_B2_ and C_C_ were relatively lower in the production diets than in the maintenance and growth diets. 

### 3.4. Gas and Methane Production Kinetics

The average values (mL/g DM) for the diets’ in vitro gas production were found to have a consistent pattern at 0–12, 12–24, and 24–48 h. The observed values for the maintenance diets were 63.0, 52.0, and 48.15; for the growth diets, they were 63.8, 52.7, and 48.2, and, for the production diets, they were 63.5, 52.5, and 47.2. The cumulative gas production values (48 h) were close and similar 163, 165, and 163 mL/g DM for the maintenance, growth, and production diets, respectively (Table 6). The in vitro methane production mean values at 0–12, 12–24, and 24–48 h and the cumulative values of the maintenance diets tended (*p* > 0.05) to be lower than those of the growth and maintenance diets, whereas, the cumulative methane production was lower in the maintenance diets (28.4) than in the production diets (33.1 mL/g DM).

### 3.5. Methane Production and Percentage Loss of Dietary Energy as Methane

The mean values of the in vitro methane production (mL/g DDM, g/kg DM and g/kg DDM) were similar in the diets formulated for maintenance (41.2, 13.3 and 29.5; Table 7), for growth (42.2, 14.3 and 30.2; Table 7), and for production (41.3, 15.9 and 29.6; Table 7). Furthermore, a similar trend was observed for the gross energy from each diet being lost as methane, with comparable values in the maintenance (1.57), growth (1.61), and production diets (1.58 kJ/g DDM), equivalent to 9.09, 9.37, and 9.14% of dietary energy lost as methane, respectively.

### 3.6. Correlation between Chemical Constituents and Methane Production 

Among the proximate constituents, the EE and lignin were significantly (*p* < 0.05) negatively associated (r = −0.422 ** and r = −0.365 **) with dietary methane production, while the OM contents of the diets were positively (*p* < 0.05 r = 0.266 *) correlated with methane production (Table 8). The protein fractions NDIP, ADIP, and P_B3_ of the diets were negatively associated with CH_4_ production (r = −0.448 **, r = −0.272 **, and r = −0.341 **). On the other hand, the N fraction, the NPN, the SP, the P_A_, and the P_B1_ fraction of the diets were positively associated (r = 0.450 **, 0.387 **, r = 0.412 **, and r = 0.284 **) with the in vitro CH_4_ production. Among the diets, the tCHO and carbohydrate fraction C_B2_ contents were positively associated (r = 0.353 ** and 0.278 **) with the in vitro CH_4_ production, while the carbohydrate fraction C_C_ DM was negatively associated (r = −0.365 **) with CH_4_ production. 

## 4. Discussion

### 4.1. Chemical Composition

All three diet categories including maintenance, growth, and production showed crude protein (CP) levels equal to or exceeding the minimum required for microbial growth. A minimum of 7.0% CP is essential to optimize the growth and functionality of rumen microbes [34]. The reason for higher CP contents in the production diets (127) than in the growth (112) and maintenance diets (82.0 g/kg DM) may be due to the inclusion of a protein-rich concentrate mixture in the production diets. These values of CP were within the range (64.4 to 150.4 g/kg DM) of values reported for 45 rations [35]. Additionally, the higher values of NDF, ADF, and cellulose contents in the maintenance diets may be due to the sole roughage ingredients in its compositions. Further, the variability in the cell wall constituents of the maintenance, growth, and production diets may be attributed to the composition and level of diverse sources of dry, green, and concentrate mixtures. The average value of the CP and hemicelluloses obtained from the growth and production diets were in similar to the value reported in a combined mixed-diet ration containing low-protein and high-protein rations [36]. The OM contents of the maintenance, growth, and production diets evaluated in the present study were within the range of OM values observed in an experiment involving seven diets [37] and utilizing local-based feed resources and tropical grass pastures [38,39]. 

### 4.2. Carbohydrate and Protein Fractions

Carbohydrates and proteins are the two most important constituents of diets required for the different physiological functions of animals, viz., maintenance, growth, and production. The total carbohydrate (tCHO) and NSC content of the maintenance, growth, and production diets recorded in the present study were aligned within the range (773.3–859.4 and 95.1–335.6 g/kg DM) of values reported for 45 rations of different roughage and concentrate feed ratios formulated using different roughage and concentrate feed types [35]. A previous study conducted on top foliage [40] and concentrate feed [10] observed values within a similar range and following a comparable trend. The maintenance, growth, and production diets had the highest contents of carbohydrate fraction C_B2_ (429, 400, and 364 g/kg DM, respectively), following a similar pattern to that of the 45 rations reported by Dong and Zhao [35]; also, our diets’ C_B2_ contents were within the range (344.8–588.2 g/kg DM) of values reported in the above-mentioned study. The variations in the concentration of the C_A_, C_B1_, C_B2_, and C_C_ carbohydrate fractions of the maintenance, growth, and production diet are similar to those observed for the 45 rations reported in the earlier study mentioned previously [35]. The carbohydrate fractions C_B2_ and C_B1_’s contents in most of our growth and production diets were within the range of values reported for six farm diets in a previous study [41]: the observed lower content of C_C_ fraction in the above-mentioned study can likely be attributed to the elevated lignin levels in the diets analyzed in our study. The higher lignin content may hinder the release or accessibility of cellulose and hemicellulose, resulting in reduced C_C_ fraction values. This phenomenon suggests a potential influence of diet composition on the structural components of plant material, with a higher lignin content acting as a limiting factor for the measured C_C_ fraction. Further, the tCHO values of the growth and production diets were similar to the tCHO values of the diets reported in the above-mentioned study [41], while their NSC contents were relatively higher than our values. The difference in the protein/nitrogen fractions of the maintenance, growth, and production/lactation diets may be attributed to the differences in the proportion of different dietary ingredients and their chemical constituents.

### 4.3. Gas and Methane Production and Loss of Energy as Methane

The average values of the gas production kinetics at three time intervals (12, 24, and 48 h) and the cumulative gas production (mL/g DM) from the high-protein and low-protein diets showed values higher than the gas production values in the maintenance, growth, and production diets in our study [36]. The gas production from the total mixed rations collected from seven dairies ranged between 211 and 256 mL/g DM after 48 h, which was higher than our gas production values. This variation in gas production may be due to differences in the chemical constituents, mainly the cell wall fractions and the carbohydrate and protein fractions, and their degradability. The availability of nutrients to microbes influences gas production from any feed/diet [42,43]. The total gas production (mL/g DM) from 45 rations of various concentrate to roughage ratios ranged between 165 and 281 [35], which partially agrees with our gas production values. The relatively higher cumulative methane production (mL/g DM) at the 48 h timepoint in the lactation diets (33.1) compared to the maintenance diets (28.4) may be due to the higher digestibility of the production diets, as the degradability of a substrate influences both gas and methane production. In a previous study conducted on lactating cows’ diets, the methane production (mL/200 g) reported was higher in the lactating ration (8.85) than in the dry ration (7.24), which substantiates our observations [44]. Further, in same study [44], they recorded higher gas production values in the lactation ration (54.4) than in the dry ration (43.0 mL/200 g). The methane production of 45 rations with varied roughage was the following: the concentrate ration ranged from 30 to 51 mL/g DM after 48 h of fermentation [35], which partially agreed with our results. A similar trend of methane production was observed in a study [45] where goats were fed three diets of different roughage to concentrate ratios (25:75, 50:50 and 73:27), and the values were 37.1, 36.4, and 34.5 g/kg DDM, respectively. Further, the CH_4_ (%GE) for these three diets (8.6, 7.3, and 6.0%) differed significantly (*p* < 0.05), which agreed with our observations that the level of concentrate and the dietary ingredients’ composition influences methane production. The percentages of CH_4_ and GE were lower in the above-mentioned study than our average values in the maintenance, growth, and production diets.

### 4.4. Correlation between Methane Production and Chemical Constituents (Proximate Constituents, Carbohydrate Fractions, and Protein Fractions)

The correlations studies between chemical constituents and CH_4_ production of forages and concentrate feeds are crucial for optimizing animal nutrition, reducing environmental impact, and improving overall feed efficiency in livestock production [34,46,47]. However, the information on the correlation between methane production and diets/rations’ chemical constituents is limited. In this study, the EE and lignin from the proximate constituents and the NDIP and ADIP protein fractions were negatively associated with methane production. Similar to our observations, earlier studies [48,49,50] reported that EE, lignin, NDIP, and ADIP were negatively associated with methane production. Contrary, a positive correlation between EE and methane production was recorded by Ellis et al. [51]. Information on diets/rations’ carbohydrate and/or protein fractions’ relationship with in vitro methane production is scarce. In a study of 45 rations, a relationship between CNCPS carbohydrate fractions and methane production was reported [35]. They also reported that the carbohydrate fractions C_A_, C_B1_, and C_B2_ were positively related to methane production, and this agrees of our correlation results. In our study, C_C_ was negatively related to methane production, and this could be due to the unavailability of lignin-bound carbohydrates for digestion. The evaluated diets’ soluble protein, NPN, P_A_, and P_B1_ were positively associated with methane production. This is probably due to the ready availability of these more degradable protein fractions to microbial fermentation.

This study effectively created diets for water buffaloes, but it has limitations. It mainly looks at diet composition and gas emissions; therefore, future research should explore the diets’ long-term effects on buffalo health, practical use on farms, and economic factors to get a fuller picture of their feasibility in real-world farming.

## 5. Conclusions

This study highlighted key findings on three categories (maintenance, growth, and production/lactation) of thirty different diet compositions for water buffaloes based on local resources, addressing the need for region-specific feeding strategies. The production diets exhibited higher crude protein contents, while the maintenance diets had more fiber. The soluble protein fractions (P_B1_ and P_B2_) were more present in the production and growth diets and the indigestible fraction (P_C_) in the maintenance diets. The higher levels of non-structural carbohydrates in the production diets suggest dietary optimization possibilities. The loss of energy as CH_4_ from the diets/feeding systems varied from 6.48% to 12.56 for the buffalos observed. Amongst the agroecological regions studied (AERs), the livestock from the AER-2 and AER-10 regions emitted the lowest CH_4_. The diets in the AER-2 and AER-10 regions consisting of tree leaves as the green fodder source produced less CH_4_, with lower losses of dietary energy as methane. The AER-10 diets supplemented with coconut cake as the protein source emitted less CH_4_. These findings emphasize the importance of tailoring diets to meet the nutritional needs of buffaloes, marking a significant step forward in optimizing buffalo farming practices. Future implications involve refining agroecological regional feeding practices and considering correlations for a targeted and sustainable diet selection process, promoting both livestock health and environmental stewardship.

## Figures and Tables

**Table 1 animals-14-00143-t001:** Proportion (%) and composition of ingredients in different concentrate mixtures.

Ingredients	CM_1_	CM_2_	CM_3_	CM_4_	CM_5_	CM_6_	CM_7_	CM_8_	CM_9_
Mustard seed cake	35	40	-	-	-	-	40	45	-
Wheat bran	25	-	25	-	25	-	-	-	-
Maize grain	40	-	-	60	-	-	20	-	40
Cotton seed cake	-	-	35	40	-	-	-	-	45
Oat grain	-	-	40	-	-	60	-	-	-
Barley grain	-	60	-	-	40	-	-	-	-
Groundnut cake	-	-	-	-	35	40	-	-	-
Rice bran	-	-	-	-	-	-	40	55	15

CM: Concentrate mixture.

**Table 2 animals-14-00143-t002:** Composition of diets (ingredients and their proportions).

AER No.	Region	Maintenance			Growth			Production		
		Diet	Composition	Proportions	Diet	Composition	Proportions	Diet	Composition	Proportions
1	Western Himalayan region	MD_1_	Grass: GOL	65:35	GD_1_	SST:L:CM_2_	60:30:10	PD_1_	WS:B:CM_2_	30:40:30
2	Eastern Himalayan region	MD_2_	Grass:LL	75:25	GD_2_	RS:LL:CM_1_	50:35:15	PD_2_	Grass: LL:CM_1_	35:40:25
3	Eastern plateau and plains region	MD_3_	RS:MG	20:80	GD_3_	RS:NG:CM_7_	30:50: 20	PD_3_	MST:NG:CM_7_	20:45:35
4	Middle Gangetic plain	MD_4_	WS:MG	50:50	GD_4_	RS:B	40:60	PD_4_	MST:CM_6_	60:40
5	Trans and Upper Gangetic plain	MD_5_	WS:B	70:30	GD_5_	SST:B:CM_2_	60:25:15	PD_5_	WS:B:CM_3_	30:40:30
6	Central plateau and hills	MD_6_	LS	100	GD_6_	GS:CM_2_	80:20	PD_6_	LS:CM_5_	60:40
7	Western plateau and hills	MD_7_	WS:SG	50:50	GD_7_	SST/L/B	55:45	PD_7_	WS:B:CM_4_	35:35:30
8	Southern plateau and hills region	MD_8_	RS:L	65:35	GD_8_	SST:ST:CM_7_	40:40:20	PD_8_	SST:CM_8_	60:40
9	Western dry zone	MD_9_	PST:LL	75:25	GD_9_	PST: LL:CM_2_	55:30:15	PD_9_	PST:CM_2_	60:40
10	Coastal and island region	MD_10_	RS:LL	65:35	GD_10_	RS:LL:CM_9_	45:40:15	PD_10_	RS:LL:CM_9_	30:35:35

AER: Agroecological regions; CM: Concentrate mixture; MD: Maintenance diet; GD: Growth diet; PD: Production diet; GOL: *Grewia optiva* leaves; LL: *Leucaena leucocephala* leaves; MG: Maize green; RS: Rice straw; SST: Sorghum stover; L: Lucerne; WS: Wheat straw; B: Berseem; LS: Lentil straw; SG: sorghum green; PST: Pearl millet stover; NG: Napier grass; MST: Maize stover.

**Table 3 animals-14-00143-t003:** Chemical composition of the maintenance diets (g/kg DM) *.

**Diet**	**CP**	**OM**	**EE**	**NDF**	**ADF**	**Cellulose**	**H cellulose**	**Lignin**
MD_1_	76.0 ^cd^	876 ^cd^	32.1 ^a^	646 ^c^	453 ^a^	298 ^c^	193 ^de^	93.4 ^b^
MD_2_	93.3 ^a^	871 ^c^	27.5 ^b^	678 ^b^	456 ^a^	274 ^d^	222 ^c^	109 ^a^
MD_3_	84.3 ^b^	923 ^a^	17.8 ^d^	668 ^bc^	361 ^d^	309 ^b^	306 ^a^	49.6 ^de^
MD_4_	69.8 ^de^	903 ^b^	14.4 ^e^	651 ^c^	379 ^c^	318 ^b^	271 ^b^	45.7 ^ef^
MD_5_	96.1 ^a^	884 ^c^	18.1 ^d^	573 ^e^	386 ^c^	301 ^c^	187 ^e^	49.7 ^de^
MD_6_	76.9 ^cd^	914 ^a^	13.4 ^e^	537 ^f^	386 ^c^	283 ^d^	151 ^f^	93.9 ^b^
MD_7_	68.0 ^e^	920 ^a^	20.7 ^cd^	713 ^a^	406 ^b^	345 ^a^	307 ^a^	42.6 ^f^
MD_8_	77.8 ^bc^	857 ^e^	26.4 ^b^	591 ^d^	391 ^bc^	272 ^d^	200 ^d^	55.8 ^cd^
MD_9_	81.8 ^bc^	920 ^a^	21.5 ^c^	545 ^f^	344 ^e^	256 ^e^	201 ^d^	62.5 ^c^
MD_10_	95.5 ^a^	852 ^e^	17.9 ^d^	575 ^e^	392 ^bc^	274 ^d^	184 ^e^	53.7 ^d^
LSD	7.12	10.20	3.17	12.96	15.62	15.07	9.17	6.84
**Diet**	**CP**	**OM**	**EE**	**NDF**	**ADF**	**Cellulose**	**H cellulose**	**Lignin**
GD_1_	121 ^ab^	917 ^ab^	19.7 ^de^	610 ^b^	411 ^b^	311 ^b^	199 ^cde^	80.7 ^b^
GD_2_	116 ^bcd^	874 ^d^	32.4 ^a^	527 ^e^	341 ^e^	231 ^c^	186 ^e^	55.4 ^fg^
GD_3_	88.7 ^e^	856 ^e^	24.7 ^bc^	676 ^a^	392 ^cd^	334 ^a^	284 ^a^	55.9 ^fg^
GD_4_	111 ^bcd^	864 ^de^	18.6 ^e^	619 ^b^	411 ^bc^	311 ^b^	209 ^cd^	52.0 ^g^
GD_5_	117 ^bc^	922 ^a^	26.7 ^b^	610 ^b^	391 ^d^	303 ^b^	219 ^c^	64.2 ^de^
GD_6_	110 ^cd^	909 ^bc^	18.4 ^e^	546 ^d^	412 ^b^	306 ^b^	134 ^f^	92.5 ^a^
GD_7_	106 ^d^	899 ^c^	17.7 ^e^	584 ^c^	389 ^d^	306 ^b^	195 ^de^	71.6 ^cd^
GD_8_	111 ^bcd^	916 ^ab^	18.1 ^e^	690 ^a^	438 ^a^	329 ^a^	252 ^b^	75.5 ^bc^
GD_9_	111 ^bcd^	917 ^ab^	23.1 ^e^	493 ^f^	300 ^f^	213 ^d^	193 ^de^	62.5 ^ef^
GD_10_	130 ^a^	872 ^d^	35.5 ^a^	512 ^e^	328 ^e^	220 ^d^	183 ^e^	52.0 ^g^
LSD	10.87	10.29	3.35	18.25	18.32	11.30	22.70	7.90
**Diet**	**CP**	**OM**	**EE**	**NDF**	**ADF**	**Cellulose**	**H cellulose**	**Lignin**
PD_1_	130 ^bc^	901 ^c^	18.9 ^de^	491 ^e^	345 ^b^	262 ^c^	146 ^f^	54.4 ^d^
PD_2_	153 ^a^	899 ^c^	39.4 ^b^	537 ^c^	292 ^d^	177 ^g^	245 ^b^	79.4 ^b^
PD_3_	103 ^e^	882 ^d^	33.6 ^cd^	633 ^a^	360 ^b^	311 ^a^	273 ^a^	45.0 ^e^
PD_4_	116 ^d^	912 ^ab^	30.7 ^e^	589 ^b^	350 ^b^	249 ^de^	239 ^bc^	68.1 ^c^
PD_5_	137 ^b^	904 ^bc^	26.3 ^f^	529 ^cd^	326 ^c^	253 ^cd^	203 ^d^	49.7 ^de^
PD_6_	121 ^cd^	918 ^a^	32.8 ^de^	453 ^f^	288 ^d^	203 ^f^	165 ^e^	74.3 ^b^
PD_7_	121 ^cd^	917 ^a^	33.4 ^cd^	549 ^c^	318 ^c^	251 ^d^	231 ^bc^	51.1 ^d^
PD_8_	121 ^cd^	902 ^c^	35.7 ^c^	634 ^a^	455 ^a^	301 ^b^	179 ^e^	98.0 ^a^
PD_9_	116 ^d^	916 ^a^	21.7 ^g^	539 ^c^	310 ^c^	240 ^e^	229 ^bc^	49.3 ^de^
PD_10_	149 ^a^	886 ^d^	49.3 ^a^	508 ^de^	283 ^d^	200 ^f^	225 ^c^	50.9 ^d^
LSD	9.82	7.95	2.68	22.11	16.41	10.41	18.67	5.68

MD, maintenance diets; GD, growth diets; PD, production diets; CP, crude protein; OM, organic matter; EE, ether extract; NDF, neutral detergent fiber; ADF, acid detergent fiber; H cellulose, hemi cellulose; LSD, least significant difference at *p* value < 0.0001; different superscript letters within a column in the table signify statistical differences among the corresponding values; *, each value is a mean of four observations.

**Table 4 animals-14-00143-t004:** Protein fractions of the diets (g/kg CP) *.

Maintenance						Growth						Production					
Diet	P_A_	P_B1_	P_B2_	P_B3_	P_C_	Diet	P_A_	P_B1_	P_B2_	P_B3_	P_C_	Diet	P_A_	P_B1_	P_B2_	P_B3_	P_C_
MD_1_	35.6 ^de^	101 ^f^	358 ^cd^	237 ^cd^	268 ^a^	GD_1_	310 ^a^	213 ^b^	228 ^f^	125 ^cd^	123 ^abc^	PD_1_	124 ^d^	242 ^bcd^	423 ^c^	95.2 ^ef^	116 ^b^
MD_2_	18.1 ^e^	95.3 ^f^	287 ^de^	301 ^b^	299 ^a^	GD_2_	27.8 ^fg^	85.4 ^d^	547 ^a^	194 ^a^	146 ^a^	PD_2_	49.4 ^g^	146 ^fg^	364 ^d^	291 ^b^	148 ^a^
MD_3_	194 ^a^	195 ^ab^	233 ^e^	157 e	220 ^b^	GD_3_	171 ^c^	213 ^b^	359 ^de^	138 ^bcd^	118 ^abcd^	PD_3_	249 ^a^	279 ^ab^	175 ^f^	187 ^c^	110 ^bc^
MD_4_	187 ^a^	185 ^bc^	327 ^d^	88.8 ^f^	211 ^bc^	GD_4_	172 ^c^	202 ^b^	385 ^cde^	155 ^b^	85.7 ^e^	PD_4_	158 ^b^	296 ^a^	304 ^e^	133 ^d^	108 ^bc^
MD_5_	173 ^a^	213 ^a^	429 ^bc^	48.9 f	135 ^def^	GD_5_	167 ^c^	295 ^a^	336 ^e^	114 ^de^	87.2 ^de^	PD_5_	134 ^cd^	259 ^abc^	464 ^e^	55.5 ^h^	87.4 ^cd^
MD_6_	123 ^b^	130 ^e^	576 ^a^	70.9 ^f^	99.6 ^f^	GD_6_	141 ^d^	292 ^a^	429 ^bc^	28.4 ^f^	110 ^bcde^	PD_6_	96.9 ^e^	208 ^ed^	508 ^bc^	113 ^de^	73.4 ^d^
MD_7_	188 ^a^	218 ^a^	24.3 ^f^	425 ^a^	145 ^de^	GD_7_	203 ^b^	192 ^b^	357 ^de^	100 ^e^	147 ^a^	PD_7_	71.3 ^f^	171 ^ef^	595 ^a^	72.1 ^fgh^	90.4 ^cd^
MD_8_	14.9 ^e^	67.8 ^g^	485 ^b^	257 ^bc^	175 ^cd^	GD_8_	109 ^e^	218 ^b^	416 ^cd^	146 ^bc^	110 ^bcde^	PD_8_	101 ^e^	223 ^cd^	492 ^b^	65.6 ^gh^	117 ^b^
MD_9_	74.4 ^c^	159 ^cd^	412 ^bc^	209 ^d^	145 ^de^	GD_9_	50.8 ^f^	188 ^b^	490 ^ab^	142 ^bc^	129 ^ab^	PD_9_	143 ^c^	228 ^cd^	431 ^c^	88.3 ^efg^	108 ^bc^
MD_10_	47.7 ^d^	138 ^de^	440 ^b^	261 ^bc^	114 ^ef^	GD_10_	15.9 ^g^	147 ^c^	550 ^a^	195 ^a^	91.4 ^cde^	PD_10_	20.2 ^h^	120 ^g^	364 ^d^	413 ^a^	82.3 ^d^
LSD	22.84	27.36	78.36	47.84	40.64	LSD	23.03	31.46	64.18	24.93	32.45	LSD	14.29	39.28	57.32	26.78	22.65

MD, maintenance diets; GD, growth diets; PD, production diets; P_A_, non-protein nitrogen; P_B1_, buffer-soluble protein; P_B2_, neutral detergent-soluble protein; P_B3_, acid detergent-soluble protein; P_C_, indigestible protein; LSD, least significant difference at *p* value < 0.0001; different superscript letters within a column in the table signify statistical differences among the corresponding values; *, each value is a mean of four observations.

**Table 5 animals-14-00143-t005:** Carbohydrate fractions of maintenance diets (g/kg DM) *.

**Diet**	**tCHO**	**NSC**	**SC**	**C_C_**	**C_B2_**	**C_B1_**	**C_A_**
MD_1_	768 ^c^	161 ^d^	607 ^c^	224 ^b^	383 ^e^	20.3 ^g^	140 ^b^
MD_2_	751 ^d^	128 ^e^	622 ^bc^	262 ^a^	360 ^f^	95.6 ^cde^	32.9 ^d^
MD_3_	821 ^ab^	185 ^c^	636 ^b^	119 ^de^	517 ^b^	74.5 ^ef^	110 ^b^
MD_4_	818 ^ab^	189 ^c^	630 ^b^	110 ^ef^	520 ^b^	87.6 ^def^	101 ^c^
MD_5_	770 ^c^	215 ^b^	555 ^d^	119 ^de^	436 ^c^	120 ^bc^	94.8 ^c^
MD_6_	824 ^ab^	300 ^a^	523 ^f^	225 ^b^	298 ^g^	178 ^a^	122 ^b^
MD_7_	832 ^a^	157 ^d^	674 ^a^	102 ^f^	572 ^a^	114 ^bcd^	43 ^d^
MD_8_	753 ^d^	196 ^c^	557 ^d^	134 ^cd^	423 ^cd^	135 ^b^	60.4 ^d^
MD_9_	817 ^b^	300 ^a^	516 ^f^	150 ^c^	366 ^ef^	58.4 ^f^	242 ^a^
MD_10_	739 ^d^	199 ^c^	540 ^e^	129 ^d^	411 ^d^	76.8 ^ef^	123 ^b^
LSD	14.23	15.14	14.70	16.43	18.56	30.53	31.46
**Diet**	**tCHO**	**NSC**	**SC**	**C_C_**	**C_B2_**	**C_B1_**	**C_A_**
GD_1_	776 ^a^	196 ^d^	580 ^b^	194 ^b^	386 ^d^	144 ^cd^	51.9 ^def^
GD_2_	726 ^c^	238 ^b^	488 ^e^	133 ^fg^	355 ^e^	175 ^bc^	62.9 ^cde^
GD_3_	743 ^b^	88.9 ^f^	654 ^a^	134 ^fg^	520 ^a^	59.2 ^g^	29.8 ^fg^
GD_4_	735 ^bc^	142 ^e^	593 ^b^	125 ^g^	468 ^b^	59.4 ^g^	82.5 ^c^
GD_5_	779 ^a^	192 ^d^	586 ^b^	154 ^de^	433 ^c^	155 ^bc^	37.3 ^efg^
GD_6_	780 ^a^	248 ^b^	532 ^d^	222 ^a^	310 ^f^	179 ^ab^	68.7 ^cd^
GD_7_	776 ^a^	218 ^c^	558 ^c^	172 ^cd^	386 ^d^	86.1 ^fg^	132 ^b^
GD_8_	786 ^a^	125 ^e^	662 ^a^	181 ^bc^	480 ^b^	108 ^ef^	16.6 ^g^
GD_9_	783 ^a^	320 ^a^	463 ^f^	150 ^ef^	313 ^f^	121 ^de^	199 ^a^
GD_10_	707 ^d^	232 ^bc^	475 ^ef^	125 ^g^	350 ^e^	208 ^a^	23.5 ^fg^
LSD	15.52	19.72	17.94	18.97	28.14	31.11	29.1
**Diet**	**tCHO**	**NSC**	**SC**	**C_C_**	**C_B2_**	**C_B1_**	**C_A_**
PD_1_	752 ^bc^	288 ^b^	464 ^d^	131 ^d^	333 ^d^	239 ^b^	49.4 ^bc^
PD_2_	707 ^d^	237 ^d^	470 ^d^	190 ^b^	280 ^f^	194 ^d^	43.1 ^c^
PD_3_	746 ^c^	143 ^f^	602 ^a^	108 ^e^	494 ^a^	103 ^g^	40.2 ^c^
PD_4_	765 ^ab^	203 ^e^	562 ^b^	163 ^c^	398 ^b^	169 ^e^	34.1 ^cd^
PD_5_	741 ^c^	231 ^e^	509 ^c^	119 ^de^	390 ^bc^	133 ^f^	98.0 ^a^
PD_6_	764 ^b^	333 ^a^	431 ^e^	178 ^b^	252 ^g^	302 ^a^	31.7 ^cd^
PD_7_	762 ^b^	233 ^d^	529 ^c^	122 ^d^	407 ^b^	217 ^bcd^	16.1 ^d^
PD_8_	745 ^c^	133 ^f^	612 ^a^	235 ^a^	376 ^c^	97.8 ^g^	35.2 ^cd^
PD_9_	779 ^a^	262 ^c^	516 ^c^	118 ^de^	398 ^b^	197 ^cd^	65.4 ^b^
PD_10_	687 ^e^	253 ^cd^	434 ^e^	122 ^d^	312 ^e^	219 ^bc^	34.1 ^cd^
LSD	13.55	22.04	22.31	13.64	19.94	24.31	19.49

MD, maintenance diets; GD, growth diets; PD, production diets; tCHO, total carbohydrates; NSC, non-structural carbohydrates; SC, structural carbohydrates; C_C_, unavailable/lignin-bound cell wall; C_B2_, slowly degradable cell wall; C_B1_, intermediately degradable starch and pectin; C_A_, rapidly degradable CHO, including sugars; LSD, least significant difference at *p* value < 0.0001; different superscript letters within a column in the table signify statistical differences among the corresponding values; *, each value is a mean of four observations.

**Table 6 animals-14-00143-t006:** Gas and methane production kinetics from the maintenance diets fermented in buffalo inoculums *.

**Diets/Rations**	**Gas (mL/g)**				**Methane (mL/g)**			
	**0–12 h**	**12–24 h**	**24–48 h**	**Cumulative**	**0–12 h**	**12–24 h**	**24–48 h**	**Cumulative**
MD_1_	64.3 ^c^	50.0 ^e^	50.0 ^b^	164 ^c^	9.83 ^d^	5.77 ^f^	5.08 ^h^	20.7 ^g^
MD_2_	58.5 ^h^	51.0 ^d^	44.3 ^f^	154 ^e^	6.57 ^f^	5.56 ^f^	11.3 ^b^	23.4 ^e^
MD_3_	59.5 ^gh^	55.8 ^b^	47.6 ^d^	163 ^c^	11.2 ^c^	12.3 ^a^	16.3 ^a^	39.8 ^a^
MD_4_	63.6 ^cd^	54.3 ^c^	50.0 ^b^	168 ^b^	12.6 ^b^	11.2 ^b^	17.0 ^a^	40.4 ^a^
MD_5_	66.0 ^b^	53.7 ^c^	45.3 ^e^	164 ^c^	11.2 ^c^	8.77 ^c^	5.66 ^g^	25.7 ^d^
MD_6_	62.3 ^de^	49.2 ^e^	47.9 ^d^	160 ^d^	15.1 ^a^	8.63 ^c^	9.56 ^d^	33.3 ^b^
MD_7_	69.8 ^a^	58.5 ^a^	48.3 ^cd^	178 ^a^	12.1 ^b^	7.89 ^d^	8.32 ^e^	28.3 ^c^
MD_8_	63.8 ^cd^	54.5 ^c^	46.1 ^e^	164 ^c^	8.72 ^e^	6.68 ^e^	7.02 ^f^	22.4 ^f^
MD_9_	61.8 ^ef^	44.5 ^g^	49.3 ^c^	156 ^e^	9.03 ^de^	6.67 ^e^	8.21 ^e^	23.9 ^e^
MD_10_	60.5 ^fg^	47.8 ^f^	52.6 ^a^	160 ^d^	7.12 ^f^	7.98 ^d^	10.8 ^c^	25.9 ^d^
LSD	1.535	1.038	0.960	2.245	0.829	0.528	0.359	1.348
**Diets/Rations**	**Gas (mL/g)**				**Methane (mL/g)**			
	**0–12 h**	**12–24 h**	**24–48 h**	**Cumulative**	**0–12 h**	**12–24 h**	**24–48 h**	**Cumulative**
GD_1_	65.0 ^b^	50.3 ^f^	48.8 ^cd^	164 ^c^	12.8 ^c^	5.74 ^h^	3.95 ^i^	22.5 ^gh^
GD_2_	59.5 ^d^	55.8 ^b^	46.8 ^e^	162 ^de^	8.20 ^ef^	10.1 ^b^	15.3 ^c^	33.6 ^c^
GD_3_	62.4 ^c^	55.2 ^bc^	49.2 ^bc^	167 ^b^	10.5 ^d^	10.6 ^a^	15.8 ^b^	36.9 ^b^
GD_4_	64.8 ^b^	52.6 ^e^	48.8 ^cd^	166 ^b^	15.5 ^a^	10.9 ^a^	16.2 ^a^	42.6 ^a^
GD_5_	65.8 ^b^	51.8 ^e^	45.6 ^f^	163 ^cd^	13.2 ^c^	7.39 ^f^	6.40 ^g^	26.96 ^f^
GD_6_	67.3 ^a^	54.5 ^cd^	44.3 ^g^	166 ^b^	10.9 ^d^	7.96 ^e^	4.73 ^h^	23.6 ^g^
GD_7_	67.8 ^a^	57.0 ^a^	48.3 ^d^	173 ^a^	14.5 ^b^	9.49 ^c^	8.22 ^f^	32.2 ^d^
GD_8_	62.8 ^c^	54.0 ^d^	49.8 ^b^	167 ^b^	9.09 ^e^	6.07 ^g^	6.40 ^g^	21.6 ^h^
GD_9_	63.5 ^c^	46.3 ^g^	49.6 ^b^	159 ^f^	11.1 ^d^	8.48 ^d^	9.67 ^e^	29.2 ^e^
GD_10_	59.5 ^d^	50.0 ^f^	51.0 ^a^	161 ^ef^	7.87 ^f^	8.50 ^d^	10.27 ^d^	26.6 ^f^
LSD	1.136	0.844	0.729	1.836	0.906	0.300	0.357	1.175
**Diets/Rations**	**Gas (mL/g)**				**Methane (mL/g)**			
	**0–12 h**	**12–24 h**	**24–48 h**	**Cumulative**	**0–12 h**	**12–24 h**	**24–48 h**	**Cumulative**
PD_1_	62.8 ^e^	54.3 ^cd^	44.0 ^g^	161 ^d^	13.9 ^b^	12.7 ^a^	14.5 ^b^	41.1 ^a^
PD_2_	62.6 ^e^	53.0 ^e^	46.8 ^e^	162 ^d^	11.1 ^d^	9.02 ^de^	14.2 ^b^	34.3 ^cd^
PD_3_	61.0 ^f^	54.6 ^bc^	48.8 ^cd^	164 ^c^	10.9 ^d^	11.2 ^bc^	16.0 ^a^	38.1 ^b^
PD_4_	64.3 ^d^	54.0 ^d^	50.0 ^ab^	168 ^b^	14.0 ^b^	11.0 ^bc^	15.9 ^a^	40.9 ^a^
PD_5_	67.6 ^a^	55.0 ^b^	45.0 ^f^	168 ^b^	15.6 ^a^	11.1 ^bc^	8.03 ^d^	34.7 ^c^
PD_6_	66.3 ^b^	54.8 ^bc^	44.1 ^g^	165 ^c^	15.0 ^a^	11.4 ^b^	6.94 ^e^	33.4 ^cd^
PD_7_	66.0 ^b^	58.6 ^a^	45.3 ^f^	170 ^a^	13.0 ^c^	10.5 ^c^	6.94 ^e^	30.4 ^e^
PD_8_	60.5 ^f^	43.8 ^h^	48.3 ^d^	153 ^f^	6.42 ^f^	5.68 ^f^	6.56 ^e^	18.6 ^g^
PD_9_	65.0 ^c^	47.5 ^g^	49.4 ^bc^	162 ^d^	13.5 ^bc^	9.41 ^d^	9.75 ^c^	32.7 ^d^
PD_10_	59.0 ^g^	49.7 ^f^	50.4 ^a^	159 ^e^	8.28 ^e^	8.63 ^e^	10.11 ^c^	27.0 ^f^
LSD	0.663	0.557	0.796	1.278	0.883	0.741	0.786	1.970

MD, maintenance diets; GD, growth diets; PD, production diets; LSD, least significant difference at *p* value < 0.0001; different superscript letters within a column in the table signify statistical differences among the corresponding values; *, each value is a mean of four observations.

**Table 7 animals-14-00143-t007:** Methane production and loss of dietary energy as methane from the maintenance diets *.

**Diets/Rations**	**IVDMD** **g/kg DM**	**CH_4_ mL/g** **DDM 24h**	**CH_4_ g/kg DM**	**CH_4_ g/kg DDM**	**GE** **kJ/g**	**GE in CH_4_ g DDM**	**CH_4_ %GE DDM**
MD_1_	422 ^bc^	37.2 ^f^	11.2 ^c^	26.7 ^ef^	16.9 ^cd^	1.42 ^ef^	8.45 ^ef^
MD_2_	402 ^c^	30.4 ^g^	8.70 ^d^	21.7 ^g^	18.0 ^ab^	1.16 ^g^	6.46 ^g^
MD_3_	482 ^b^	48.9 ^a^	16.9 ^a^	35.0 ^a^	17.5 ^bc^	1.87 ^a^	10.7 ^ab^
MD_4_	468 ^b^	49.4 ^a^	17.1 ^a^	35.4 ^a^	16.3 ^d^	1.89 ^a^	11.6 ^a^
MD_5_	468 ^b^	42.5 ^bcd^	14.3 ^b^	30.5 ^bcd^	17.5 ^bc^	1.63 ^bcd^	9.31 ^cde^
MD_6_	584 ^a^	43.2 ^bc^	17.0 ^a^	31.0 ^bc^	18.7 ^a^	1.66 ^bc^	8.87 ^de^
MD_7_	417 ^bc^	47.0 ^ab^	14.4 ^b^	33.7 ^ab^	17.4 ^bc^	1.80 ^ab^	10.3 ^bc^
MD_8_	367 ^e^	41.5 ^cde^	11.0 ^c^	29.8 ^cde^	16.2 ^d^	1.59 ^cde^	9.8 ^bcd^
MD_9_	474 ^bc^	33.5 ^fg^	11.3 ^c^	24.1 ^fg^	17.6 ^bc^	1.28 ^fg^	7.31 ^fg^
MD_10_	389 ^de^	38.5 ^de^	10.8 ^c^	27.6 ^de^	17.2 ^c^	1.47 ^de^	8.56 ^e^
LSD	43.58	4.64	0.81	3.33	0.768	0.178	1.14
**Diets/Rations**	**IVDMD** **g/kg DM**	**CH_4_ mL/g** **DDM 24 h**	**CH_4_ g/kg DM**	**CH_4_ g/kg DDM**	**GE** **kJ/g**	**GE in CH_4_ g DDM**	**CH_4_ %GE DDM**
GD_1_	450 ^d^	41.3 ^bc^	13.27 ^e^	29.6 ^bc^	16.9 ^cd^	1.57 ^bc^	9.29 ^bc^
GD_2_	496 ^bc^	37.0 ^de^	13.14 ^e^	26.5 ^de^	17.5 ^bc^	1.41 ^de^	8.06 ^e^
GD_3_	530 ^b^	39.9 ^cd^	15.13 ^c^	28.6 ^cd^	16.9 ^cd^	1.53 ^cd^	9.07 ^cd^
GD_4_	628 ^a^	42.1 ^bc^	18.94 ^a^	30.2 ^bc^	17.0 ^cd^	1.61 ^bc^	9.49 ^bc^
GD_5_	466 ^cd^	44.3 ^b^	14.74 ^c^	31.8 ^b^	16.9 ^cd^	1.7 ^b^	10.0 ^b^
GD_6_	530 ^b^	35.7 ^e^	13.55 ^de^	25.6 ^e^	16.6 ^d^	1.37 ^e^	8.21 ^de^
GD_7_	409 ^e^	58.7 ^a^	17.21 ^b^	42.1 ^a^	17.9 ^ab^	2.24 ^a^	12.5 ^a^
GD_8_	372 ^f^	40.9 ^bc^	10.87 ^g^	29.3 ^bc^	16.7 ^d^	1.57 ^bc^	9.43 ^bc^
GD_9_	469 ^cd^	41.9 ^bc^	14.01 ^d^	30.0 ^bc^	18.3 ^a^	1.61 ^bc^	8.82 ^cde^
GD_10_	408 ^e^	40.1 ^cd^	11.74 ^f^	28.8 ^cd^	17.4 ^bc^	1.53 ^cd^	8.83 ^cde^
LSD	35.36	3.88	0.686	2.78	0.614	0.148	0.865
**Diets/Rations**	**IVDMD** **g/kg DM**	**CH_4_ mL/g** **DDM 24h**	**CH_4_ g/kg DM**	**CH_4_ g/kg DDM**	**GE** **kJ/g**	**GE in CH_4_ g DDM**	**CH_4_ %GE DDM**
PD_1_	621 ^a^	42.9 ^c^	19.06 ^a^	30.7 ^c^	17.5 ^bcd^	1.66 ^c^	9.48 ^cd^
PD_2_	605 ^a^	33.2 ^de^	14.39 ^d^	23.8 ^de^	17.7 ^abc^	1.28 ^de^	7.24 ^e^
PD_3_	600 ^ab^	36.9 ^d^	15.82 ^c^	26.5 ^d^	16.8 ^ef^	1.41 ^d^	8.40 ^d^
PD_4_	523 ^cd^	47.9 ^ab^	17.92 ^b^	34.3 ^ab^	16.9 ^def^	1.82 ^ab^	10.8 ^ab^
PD_5_	520 ^d^	51.5 ^a^	19.11 ^a^	36.9 ^a^	17.4 ^bcd^	1.99 ^a^	11.4 ^a^
PD_6_	563 ^bc^	47.0 ^bc^	18.96 ^a^	33.7 ^ab^	16.7 ^f^	1.78 ^bc^	10.7 ^ab^
PD_7_	496 ^d^	47.4 ^ab^	16.82 ^c^	34.0 ^ab^	18.1 ^a^	1.82 ^ab^	10.1 ^bc^
PD_8_	391 ^e^	31.1 ^e^	8.68 ^f^	22.3 ^e^	17.2 ^cdef^	1.20 ^e^	6.92 ^e^
PD_9_	534 ^cd^	43.1 ^c^	16.4 ^c^	30.9 ^c^	17.3 ^bcde^	1.66 ^c^	9.59 ^c^
PD_10_	533 ^cd^	31.8 ^e^	12.1 ^e^	22.8 e	17.8 ^ab^	1.20 ^e^	6.84 ^e^
LSD	40.91	4.21	1.003	3.02	0.575	0.161	1.029

MD, maintenance diets; GD, growth diets; PD, production diets; IVDM, in vitro dry matter digestibility; GE, gross energy; LSD, least significant difference at *p* value < 0.0001; different superscript letters within a column in the table signify statistical differences among the corresponding values; *, each value is a mean of four observations.

**Table 8 animals-14-00143-t008:** Correlation between in vitro methane production and chemical constituents of the diets/rations.

Chemical Constituents	CH_4_ g/g DDM	Protein Fractions	CH_4_ g/g DDM	CHO Fractions	CH_4_ g/g DDM
CP	−0.134	NDIP	−0.448 (**)	tCHO	0.353 (**)
OM	0.266 (**)	ADIP	−0.272 (**)	NSC	0.115
EE	−0.422 (**)	SP	0.387 (**)	SC	0.083
NDF	−0.009	NPN	0.450 (**)	Starch % NSC	−0.104
ADF	−0.127	P_A_	0.412 (**)	C_C_ DM	−0.365 (**)
Cellulose	−0.073	P_B1_	0.284 (**)	C_B2_DM	0.278 (**)
Hemi cellulose	0.130	P_B2_	−0.053	C_B1_DM	0.031
Lignin	−0.365 (**)	P_B3_	−0.341 (**)	C_A_DM	0.091
Energy	−0.032	P_C_	−0.145		

CP, crude protein; OM, organic matter; EE, ether extract; NDF, neutral detergent fiber; ADF, acid detergent fiber; NDIP, neutral detergent-insoluble protein; ADIP, acid detergent-insoluble protein; SP, soluble protein; P_A_, non-protein nitrogen; P_B1_, buffer-soluble protein; P_B2_, neutral detergent-soluble protein; P_B3_, acid detergent-soluble protein; P_C_, indigestible protein; tCHO, total carbohydrates; NSC, non-structural carbohydrates; SC, structural carbohydrates; C_C_, unavailable/lignin-bound cell wall; C_B2_, slowly degradable cell wall; C_B1_, intermediately degradable starch and pectin; C_A_, rapidly degradable CHO, including sugars; DM, dry matter; DDM, digestible dry matter; **, statistically significant.

## Data Availability

Data are contained within the article.

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
