# Peer review of "Agroecological Zone-Specific Diet Optimization for Water Buffalo (Bubalus bubalis) through Nutritional and In Vitro Fermentation Studies"

_animals, 2023, doi:10.3390/ani14010143_

Round 1

Reviewer 1 Report

Comments and Suggestions for Authors

1.       The analytical methodology is not sufficiently specific and needs to be further refined.(2.9)

2.       Why only two fistulated adult male water buffaloes were used?

3.       Table 3 could become Table 3.1, Table 3.2, Table 3.3. Table 5 could become Table 5.1, Table 5.2, Table 5.3.

4.       Problems with the content of the table notes. “values with superscripted letters indicate significant differences in the values; *,”

5.       The Results and Discussion section mostly expresses comparisons between the three phase diets, but the authors set 10 in the same phase diets and should not ignore within-group comparisons. Then your analysis method may need to be modified.

6.       In the conclusion section of the paper, the description is too formalized and does not summarize and give insights for the results of the experiment. For example, through interaction analysis the authors found a correlation between methane production and the chemical composition of the diets, so which diet at different stages had chemical compositions consistent with low methane production? In the case of different dietary ingredients, one can also configure new diets based on the chemical composition characteristic of low methane production.

Author Response

Dear Reviewer,

We thank you for your highly insightful comments and the time devoted to this manuscript. It significantly helped me in the improvisation of the quality of our article. I have incorporated all comments and advice in the main file of the revised manuscript. In the attached file, I have written the responses point by point for each comment.

Regards,

Reviewer 2 Report

Comments and Suggestions for Authors

Brief summary: The paper entitled Agroecological Zone-Specific Diets Optimization for Water  Buffalo (Bubalus bubalis) through Nutritional and In Vitro Fermentation Studies' investigate 30 different diets, using local food ingredients, divided into three groups of 10, each meeting the maintenance (MD1 to MD10), growth (GD1 to GD10) and lactation/production (PD1 to PD10) needs of the buffaloes. Using the Cornell system, the carbohydrate and protein contents of the various diets were evaluated. The production diets had more protein and fat than the maintenance diets. Fibre was lower (p<0.05) in the production diets than in the maintenance diets. Several protein components (PB1, PB2) were lower (p<0.05) in maintenance diets than in growth and production diets, whereas other protein fractions (PB3, Pc) were higher (p<0.05) in maintenance diets. The carbohydrate fraction (CB1) was highest in the production diets, followed by the growth and maintenance diets.  In contrast, the CA carbohydrate fraction was higher in maintenance diets than in growth diets and production diets. In vitro gas production over time (12, 24 and 48 hours) was more or less the same for all diets.

The authors conclude that methane was produced less by the maintenance diets than by the growth and production diets, although the amount of methane relative to the amount of digested matter was virtually the same for all diets. the different diets had different nutritional components and this affected the gases produced during diet digestion.

The paper is interesting and in line with the aim of the journal and the special issue, but in its current form it is not publishable, in fact many additions are needed to improve it.

Here are my observations, line by line

Line 61: Check "(14.08Tg) [5]."

Lines 75-90: the objective of the study should be clearly specified.

Lines 186-187: 2.9. Statistical analysis. The analysis of the data was conducted utilising the SAS software, specifically employing the general linear model (GLM) procedure. The model used must be defined, including the modelled variables, correlations, significance, the tests used, and the expression of the results.

Table 2. row: "Eastern plateau and plains region" check composition (7). In the penultimate column "Composition", I did not find the CM6. 

Line 219. Table 3.1, check NDF 6467 and verify is the corresponding LSD value (12.96) is correct; check also the lignin value 52.0 g

Line 244: Table 5. 1:. Check MD8 CP "60.d4".

Line 256: Table 6. 1:. Check MD4 "17..0a".

Line 261: Table 6. 2. Why did the authors only report the p-value in this table?

Line 338: Similar pattern of 45 rations reported by Dong and Zhao et al [34]. Rewrite as similar pattern of 45 rations reported by Dong and Zhao [34].

Lines 386-388: In this study, EE and lignin proximal components and protein fractions NDIP and ADIP were negatively associated with methane production (r = -0.422** and r 387 = - 0.365**) and (r = -0.448** and r = -0.272**).

In the discussion it is not necessary to give the 'r values' because they are already given in the table and the results. give the 'r values' because they are already given in the table and the results.

I suggest extending the discussion and rewriting the conclusions more clearly.

Comments on the Quality of English Language

I am of the opinion that a moderate amount of editing of the English language is necessary.

Author Response

(The authors gave the same response as above.)

Reviewer 3 Report

Comments and Suggestions for Authors

Abstract:

1.    This abstract lack background information and fails to clearly state the problems the study aims to address and its significance. In addition, the results are redundant, lacking emphasis on meaningful findings. And the conclusion section in the abstract does not effectively highlight the significance of the study. 

Introduction:

2.     This sentence is not relevant to the study and is recommended to be deleted. The entire abstract section is too cumbersome to properly elicit the purpose of this study. (Line 58-59)

3.     Add the reason and basis of choosing different diets. The stages of maintenance, growth and lactation/production should be defined, and the differences in nutrient requirements at the three stages should be briefly described. (Line 69-70)

Materials and methods

4.     Complement the reference of design concentrate mixtures. (Line 94-96)

5.     Provide rationale for not using heat-stable α-amylase and sodium sulphite in NDF determination. (Line 111-112)

6.     Clarification of whether sex had an effect on experimental results should be supplemented. (Line 142-144)

Results and Discussion:

7.     This is a missing word in this sentence before (82.0 and 21.0 g/kg DM). (line 192)

8.     There is no statistically significantly difference among in vitro methane production at various time intervals (0-12, 12-24, 24-48 hours) among different diets. Please provide the exact p value in aid in the interpretation of these findings. (line 239-243)

9.     [YW2] In the results section, the description of experimental data should be supplemented to highlight the research focus. (Table3-8)

10.  Limitability of the experiment design and further efforts of the study should be discussed.

11.  The analysis is incomplete for the specific diets (MD, GD, and PD) within each Agroecological Region (AER). The average values for MD, GD, and PD across all AERs may not accurately represent the data. Please share additional details or clarify why this information is missing.

Conclusions

12.  In this paper, the same rumen fluid samples were used in the in vitro samples of the three stages, which lacked repeatability and rigor and could not draw a conclusion. (Line 409-413)

References:

13.  Please recheck and use the uniform format for all references.

Comments on the Quality of English Language

There are some problems in this article, such as improper use of definite articles and logical confusion. It is suggested to seek the help of a professional to revise the writing. 

Author Response

(The authors gave the same response as above.)

Round 2

Reviewer 1 Report

Comments and Suggestions for Authors

If your aim is to compare the differences between diets at different growth stages, then the format in which your results are presented needs to be changed. If it is presented in its current form, then you should add a comparison of diets from the same time period.

Author Response

Dear Reviewer,

We thank you again for your suggestion and the time devoted to our manuscript.

Regarding your comment;

we have our explanation that 

Our purpose was to evaluate the diets formulated for different stages (maintenance, growth and production) based on local feed resources for their variability in nutritional quality and methane emission so that low methane production diets may be formulated in different regions.

Reviewer 2 Report

Comments and Suggestions for Authors

The paper has been greatly improved, I just have one small suggestion, check this sentence:

"Therefore, the primary objective of this work was to formulate and assess 30 water buffalo diets, designed for different life stages and agroecological zones in India, evaluating their nutritional compositions, in vitro methane production, and ultimately redirecting methane emissions into a valuable energy source to enhance liv-stock productivity while addressing global environmental concerns.

Congratulations to the authors for their work

Comments on the Quality of English Language

Minor editing of English language required

Author Response

Dear Reviewer,

We thank you again for your suggestions and the time devoted to this manuscript. Below, I have written the response to your point for the mentioned comment.

The paper has been greatly improved, I just have one small suggestion, check this sentence:

"Therefore, the primary objective of this work was to formulate and assess 30 water buffalo diets, designed for different life stages and agroecological zones in India, evaluating their nutritional compositions, in vitro methane production, and ultimately redirecting methane emissions into a valuable energy source to enhance liv-stock productivity while addressing global environmental concerns.

Response: Sentence checked and rephrased as per the suggestion

Reviewer 3 Report

Comments and Suggestions for Authors

Abstract:

1.  The originality and significance of content is not clear. 

2.  There is no statistically significant difference in the gas production among all diet. It is not appropriated to state that different diets have an impact on gases production. (line 45)

Introduction:

3. Please provide contents regarding current buffalo diets and specify the challenges faced in buffalo farming.

Results and Discussion:

4.  The superscripts on the table are not annotated. (table 3-7)

Conclusions

5.   Similar to the abstract, please provide additional details regarding the originality and significance of content. 

Comments on the Quality of English Language

There are some problems in this article, such as improper use of words and logical confusion. It is suggested to seek the help of a professional to revise the writing. 

Author Response

Dear Reviewer,

We thank you again for your suggestions and the time devoted to this manuscript. Below, I have written the responses point by point for each comment. Please have a look at below:

Abstract:

  1. The originality and significance of content is not clear. 

Response: Information is added as per the suggestion.

  1. There is no statistically significant difference in the gas production among all diet. It is not appropriated to state that different diets have an impact on gases production. (line 45)

Response: Thanks for notifying content was removed as per the suggestion.

Introduction:

  1. Please provide contents regarding current buffalo diets and specify the challenges faced in buffalo farming.

Response: The related content is added in the introduction section.

Results and Discussion:

  1. The superscripts on the table are not annotated. (table 3-7)

Response: The superscript letters within a column in the table indicate that values sharing the same letter are not statistically different from each other, while values with different superscript letters are statistically distinct and the same is added in the all the tables.

Conclusions

  1. Similar to the abstract, please provide additional details regarding the originality and significance of content. 

Response: The related content is added in the conclusion section as per the suggestion.